# A Cancer Spheroid Array Chip for Selecting Effective Drug

**DOI:** 10.3390/mi10100688

**Published:** 2019-10-12

**Authors:** Jae Won Choi, Sang-Yun Lee, Dong Woo Lee

**Affiliations:** 1Department of Biomedical Engineering, Konyang University, Daejeon 35365, Korea; zeak3659@naver.com; 2Department of Health Sciences and Technology, SAIHST, Sungkyunkwan University, Seoul 06351, Korea; leesangyun316@gmail.com; 3Medical & Bio Device (MBD), Suwon 16229, Korea

**Keywords:** organoid, 3D cell culture, spheroid array, high-throughput screening, drug efficacy

## Abstract

A cancer spheroid array chip was developed by modifying a micropillar and microwell structure to improve the evaluation of drugs targeting specific mutations such as phosphor-epidermal growth factor receptor (p-EGFR). The chip encapsulated cells in alginate and allowed cancer cells to grow for over seven days to form cancer spheroids. However, reagents or media used to screen drugs in a high-density spheroid array had to be replaced very carefully, and this was a tedious task. Particularly, the immunostaining of cancer spheroids required numerous steps to replace many of the reagents used for drug evaluation. To solve this problem, we adapted a micropillar and microwell structure to a spheroid array. Thus, culturing cancer spheroids in alginate spots attached to the micropillar allowed us to replace the reagents in the microwell chip with a single fill of fresh medium, without damaging the cancer spheroids. In this study, a cancer spheroid array was made from a p-EGFR-overexpressing cell line (A549 lung cancer cell line). In a 12 by 36 column array chip (25 mm by 75 mm), the spheroid over 100 µm in diameter started to form at day seven and p-EGFR was also considerably overexpressed. The array was used for p-EGFR inhibition and cell viability measurement against seventy drugs, including ten EGFR-targeting drugs. By comparing drug response in the spheroid array (spheroid model) with that in the single-cell model, we demonstrated that the two models showed different responses and that the spheroid model might be more resistant to some drugs, thus narrowing the choice of drug candidates.

## 1. Introduction

When using conventional approaches for evaluating anticancer drugs, the 2-dimensional monolayer (2D) cell culture model is the gold standard. However, when cancer cells are cultured in plastic dishes, the cell morphology differs from the 3D growth occurring in animal cells in the living body. This environment also affects gene expression. It has been reported that animal cells grown in biocompatible 3D cell culture models exhibit different gene expression patterns than when they are grown in 2D cell culture models [1]. As a result, in vitro animal cell cultures have poor correspondence with in vivo animal cell cultures. Thus, many 3D cell culture models have been developed to overcome this poor correspondence [2]. Moreover, when animal cells are used for analyzing drug efficacy or toxicity, the drug reactivity in a 3D cell culture model differs greatly from what has been observed in conventional 2D cell culture models [3,4,5,6]. When cells derived from cancer patients are cultured in 3D, cell-cell interactions and the extracellular matrix (ECM) change the morphology of the cells in the culture, as well as the type and expression level of the major genes being expressed [3,4,5,6]. For these reasons, tools aiding in the development of 3D cell cultures are being studied, and some have even been commercialized. Generally, a 3D cell culture can be categorized into two models. A scaffold-free model allows cells to grow together without exogenous extracellular matrix and a scaffolding model allows cell cultivation in the ECM space. Recently, the scaffold method was used to form cancer organoids (spheroids over 100 µm in diameter), which are considered as near-physiological in vitro cell models [7,8,9]. Organoids could be used for biomedical research, genomic analysis of various diseases, and therapeutic studies [10,11,12,13,14,15,16,17]. Specifically, cancer organoid cultures could be a powerful tool for evaluating drug efficacy and toxicity during drug discovery studies [18], for conducting cytotoxicity investigations of new therapeutic compounds [19], as well as for personalizing cancer treatments [9,20]. Thus, many technologies such as hang drop technology [21], agarose microwells [22], and microfluidic chips [23] have been developed, which successfully demonstrate the performance of cancer organoid cultures. However, for commercial application of high throughput screening, the issue of automation needs to be resolved. Especially, while screening drugs in a high-density cancer spheroid array, the media need to be changed by careful pipetting, which is a tedious task, and a bottleneck in automation. To overcome this problem, we adopted a micropillar and microwell structure of the spheroid array, as shown in Figure 1. Previously, we have described a micropillar and microwell chip for culturing 3D cells and testing drug efficacy [24,25,26]; however, the drugs were exposed to single cells or small spheroids.

In this study, as shown by the spot images following the number of days in Figure 1, cancer spheroids over 100 µm in diameter began to form on a micropillar chip at day seven. After day seven, cancer spheroids maintained their size and overexpression of phosphor-epidermal growth factor receptor (p-EGFR). Culturing cancer spheroids in alginate spots attached to the micropillar allowed the media to be changed by replacing with new microwells filled with fresh media. Sixteen cancer cell lines successfully grew and formed spheroids over 100 µm in diameter in alginate spots on the micropillar chip, as shown in Figure 1b. We applied the cancer spheroid array to identify highly effective drugs targeting p-EGFR from among seventy drugs (including ten EGFR-targeting drugs as model drugs) by using a p-EGFR-overexpressing cell line (A549 cell line). The micropillar and microwell chip could support the single-cell and spheroid models and produced different drug responses according to the model.

## 2. Materials and Methods

### 2.1. Fabrication of Micropillar/Microwell Chips and Incubation Chamber for the Spheroid Array

The micropillar/microwell chip platform is typically used for short-term cell cultures to form spheroids. However, a technical issue arose when we applied this technique for long-term cell culture to form spheroids; the tightly combined chips lacked a CO_2_ supply. Moreover, the low volume (1 µL) of media in the microwell evaporated very easily. To resolve these issues, the micropillar/microwell chips were modified; the micropillar was combined with the microwell chip very tightly to prevent evaporation. Nonetheless, this caused the combined chips to run out of CO_2_ at a higher rate. Adding a spacer to the microwell chip (Figure 2c) created a gap between the chips so that CO_2_ could easily penetrate the wells. The modified micropillar and microwell chips were manufactured by plastic injection molding. The micropillar chip was made of poly styrene-*co*-maleic anhydride (PS-MA) and contained 532 micropillars (with a 0.75 mm pillar diameter and a 1.5-mm pillar-to-pillar distance). PS-MA provides a reactive functionality to covalently attach poly-L-lysine (PLL), ultimately attaching alginate spots by their ionic interactions. Plastic molding was performed with an injection machine (Sodic Plustech Inc., Schaumburg, IL, USA). To prevent evaporation, an incubation chamber for micropillar/microwell chips (Figure 2d) was fabricated by cutting the cyclic olefin copolymer (COC) with a computer numerical control (CNC) machine. The COC was selected because of its high transparency, excellent biocompatibility, and adequate stiffness for physical machining. As shown in Figure 2d, four combined chips were placed in the reservoir, which was filled with distilled water (DI) to prevent evaporation. After 13 days, 5.3% of the media had evaporated from the incubation chamber.

### 2.2. Cell Line Culture

A549 (lung), SK_GT_4 (esophagus), MKN_1 (stomach), OE19 (esophagus), SNU-638 (stomach), SNU-719 (stomach), SNU-601 (stomach), AGS (stomach), KYAE-1 (esophagus), H82 (lung), Hep3B (liver), SW48 (colorectal), MKN_45 (stomach), KATOIII (stomach), SNU-520 (stomach), and ESO26 (adenocarcinoma of the gastroesophageal junction) were cultured in RPMI 1640 medium (Gibco, Co Dublin, Ireland) supplemented with 10% fetal bovine serum (FBS) and 1% antibiotics (Gibco, Co Dublin, Ireland). All cell lines were purchased from the Korean Cell Line Bank (Seoul, South Korea). Cell lines were maintained at 37 °C in a 5% CO_2_, in a humidified atmosphere and passaged every four days. Normally, we used cell lines under 20 passages after thawing the frozen cell stock. Under 20 passages, sixteen cell lines could easily form 3D cells in 0.5% (w/w) alginate on the chip platform.

### 2.3. Experimental Procedure

Approximately 100 cells in 50 nL of 0.5% (w/w) alginate were automatically dispensed onto a micropillar chip by using an ASFA™ Spotter ST (Medical & Bio Decision, Suwon, South Korea). ASFA™ Spotter ST uses a solenoid valve (The Lee Company, Westbrook, CT, USA) for dispensing the 50-nL droplets of the cell–alginate mixture and 1 µL of medium or drugs. The top of the micropillar was coated with 60 nL of 0.02 M BaCl_2_. When the cell-alginate mixture was dispensed on the top of the micropillar, the barium ions replaced the sodium ions in the alginate; thus, the polymer strands in the alginate was cross-linked with the barium ions, resulting in an alginate gel. After dispensing the cells and media in the micropillar and microwell, respectively (Figure 2a), the micropillar chip containing the human cells in the alginate was combined (or “stamped”) with the microwell chip filled with 1 µL of fresh media (Figure 2b). The micropillar and microwell chip in their combined form is shown in Figure 2b. After three days of incubation at 37 ℃ to stabilize the cells, the cells started forming spheres. We changed the media and allowed cells to grow up to seven days until the size of the spheres was larger than 100 µm. In order to uniformly infiltrate the CO_2_ into the chip, the micropillar chips were spaced 200 µm apart from the microwell chip, as shown in Figure 2c. While incubating the cells on the chip, an incubation chamber (Figure 2d) was used to prevent evaporation of the media, even though the spacer made evaporation easier. On day seven, the drug was added to the spheroids over 100 µm in diameter, which were now cancer organoids Figure 2e. To evaluate the cancer spheroids, immunostaining was performed following the protocol described in the previous work [19]. Briefly, 3D cells cultured on the micropillar chip were fixed using a 4% paraformaldehyde solution (PFA, Biosesang, Seongnam, Korea) mixed with 2.5 mM CaCl_2_ for 120 min. The amount of time required to fix 3D-cultured cells is considerably longer than the 60 min required to fix a 2D cultured cell. Experimentally, 120 min is the minimum amount of time required to prevent cell degradation in alginate. After fixation, the micropillars were transferred to a permeabilizing and blocking solution (1% bovine serum albumin (BSA) in phosphate-buffered saline (PBS) containing 0.3% Triton-X) for 1 h. Subsequently, each micropillar chip was incubated overnight at 4 °C with the antibody staining solution. The antibody staining solution was prepared by adding anti-p-EGFR (200:1, Abcam, Cambridge, UK, phosphor Y1092, Alexa Fluor 488, green fluorescent dye with excitation at 488-nm), Hoechst 33342 (1000:1, Thermo Fisher Scientific, Waltham, MA, USA, Hoechst 33342, blue fluorescent dye that could be excited using a 358-nm laser), and F-actin phalloidin (400:1, Thermo Fisher Scientific, Waltham, MA, USA, Alexa Fluor 594, red fluorescent dye that could be excited using 561-nm or 594-nm lasers) to the permeabilizing and blocking solution. The stained chip was then washed for 15 min in the staining buffer solution (MBD-STA500, Medical & Bio Device, Suwon, South Korea) and then dried completely in a dark environment. To image the stained cells, the micropillars were scanned using an optical scanner (ASFA™ Scanner HE, Medical & Bio Device, Suwon, Korea).

To evaluate the efficacy of 70 different drugs in inhibition of p-EGFR using a single chip, we dispensed 70 drugs at a concentration of 20 µM into the microwell. One alginate spot containing cells on a micropillar from among the 12 by 36 array was exposed to one drug (20 µM). One microliter of each drug was dispensed into a microwell using the ASFA™ Spotter ST (Medical & Bio Decision, Suwon, South Korea). One microliter of each drug could be dispensed using Echo Liquid Handlers (LABCYTE, San Jose, CA, USA). Each drug was administered to the six microwells (six replicates). In our previous study [27,28], even though cells were treated with drugs for one day (at almost a single-cell stage, before forming spheroids), most drugs showed high resistance in 3D-cultured cells. Thus, a high dose of 20 µM of drug was selected because the spheroids over 100 μm in diameter may show high resistance to drugs. The purpose of this paper was to demonstrate that, because of the high drug resistance, spheroids over 100 µm in diameter can help narrow down the options for drug candidates targeting p-EGFR. We expected the spheroids to have a higher resistance to drugs than the 3D single-cell model; therefore, we chose 20 µM as the drug concentration.

### 2.4. Comparison of Drug Response between the Single-Cell and Spheroid Models

To determine the differences in results obtained with each model, we compared the responses to drug treatment. For the single-cell response, cells were treated with drugs on day one, and cell viability was measured on day seven. For the spheroid response, spheroids were treated with drugs on day seven, and spheroid viability was measured on day 13, as shown in Figure 2. The drug treatment time and p-EGFR staining times (6, 24, 48, and 144 h) in the spheroid model were similar to those in the single-cell model. To compare p-EGFR expression and cell viability for the different chips in different days, we normalized cell viability and p-EGFR expression with those in the control (no drugs) in the same chip. Thus, p-EGFR was normalized by dividing its relative expression in the control sample with that in a drug treatment sample on the same chip. Chips 6, 24, 48, and 144 h were stained after administering the drug and normalized p-EGFR at each time point because p-EGFR inhibition occurred at different times depending on the drugs. Among the normalized p-EGFR samples taken at 6, 24, 48, and 144 h, a minimum p-EGFR normalization value was selected for evaluating drug inhibition of p-EGFR.

### 2.5. p-EGFR Measurement

The nucleus, p-EGFR, and F-actin in the 3D-cultured cells in alginate were stained with different color fluorescence (blue, green, and red). After fixing the 3D-cultured cells, the red fluorescent dye was used for identifying filamentous actin (F-actin)—one of the components of the cytoskeleton—because the cytoskeleton degrades when the 3D-cultured cells are affected by the drugs. The green fluorescent dye was used to identify p-EGFR in the cell membrane, whereas the blue fluorescent dye was used to identify the cell nucleus. An automatic optical fluorescence scanner (ASFA™ Scanner ST, Medical & Bio Device, Suwon, South Korea) was used to measure the red, green, and blue fluorescence intensities using an 8-bit code among the RGB codes (0–255); the 3D-cultured cells were identified according to intensity thresholds (20 green code). The relative p-EGFR expression levels (relative p-EGFR) were calculated by dividing the green area (p-EGFR) with the blue area (the nucleus of the cell) in one alginate spot, as shown below:
(1)$$\mathrm{Relative}\;p-\mathrm{EGFR}\;\left[\%\right]=\frac{Total\;Green\;Area}{Total\;Blue\;Area}\times 100$$

The cytoskeleton was measured by calculating the size of the red area (F-actin expression). F-actin and p-EGFR values were normalized to their corresponding controls (no drug treatment).

### 2.6. Viability Measurement

Calcein AM staining solution was prepared by adding 1.0 µL of calcein AM (4 mM stock from Invitrogen) in 8 mL of 140 mM NaCl supplemented with 20 mM CaCl_2_. The non-fluorescent acetomethoxy derivate of calcein (calcein AM, AM = acetoxymethyl) is used because it can be transported through the cellular membrane into live cells. After transport into the cells, intracellular esterases remove the acetomethoxy group, the molecule gets trapped inside, and gives out strong green fluorescence. As dead cells lack active esterases, only live cells are labeled. 

To check cell viability at day seven after drug treatment in the single and spheroid models, cells in the chips were stained with Calcein AM. The live cells were stained and produced green fluorescence. An automatic optical fluorescence scanner (ASFA™ Scanner ST, Medical & Bio Device, Suwon, South Korea) was used to measure green fluorescence intensities using an 8-bit code among the RGB codes (0–255). The area of the 3D-cultured cells was identified according to the intensity threshold (20 green codes) to reduce the background noise. The green area was used for calculating the cell viability. To determine relative viability, the green area for the cells exposed to the drug was divided by the control cell area without drug exposure. The relative viabilities are based on healthy cells without drug exposure. Six alginate spots were used for calculating the average and standard deviation of the relative viability, shown in Table 1.

### 2.7. Western Blot Assay

Total cell lysates from the A549 lung cancer cell lines were prepared using the Complete™ Lysis-M buffer solution (Roche Life Science, Penzberg, Germany). Protein extracts were resolved using 4–20% Mini-PROTEAN TGX™ Precast Protein Gels (Bio-Rad, Berkeley, CA, USA) and transferred onto iBlot® PVDF gel transfer stack membranes (Thermo Fisher Scientific, Waltham, MA, USA). After blocking non-specific binding sites for 1 h in 5% BSA in Tris-buffered saline containing 0.1% Tween-20 (TBS-T), the membranes were incubated overnight at 4 °C with specific primary antibodies. The antibodies included anti-p-EGFR (phospho Y1092) antibody (1:1000, Abcam, Cambridge, UK) and anti-beta actin (1:2000, Abcam, Cambridge, UK). These were used following the manufacturers’ instructions.

## 3. Results and Discussion

3D cell cultures such as spheroids could be a highly useful tool for simulating the cancer microenvironment and predicting drug efficacy. To precisely predict the in vivo efficacy of targeting drugs, a spheroid cytoskeleton, as well as the expression levels of a target protein (p-EGFR), were measured by immunofluorescence staining in a high throughput manner. Cell viabilities were also measured by Calcein AM staining in a different chip. Using micropillar and microwell chips, the changes in expression levels of p-EGFR were measured in 3D-cultured cells to screen the targeting efficiency of 70 drugs. By comparing p-EGFR expression levels and cell viability in the single-cell model with those in the spheroid model, the possible drug candidate options could be narrowed down. Before drug administration, immunostaining for p-EGFR using the chip was verified by measuring p-EGFR in AGS [29] and A549 [30], which are well-known cell lines of p-EGFR overexpression. Figure 3 shows the relative expression levels of p-EGFR in these two cell lines. The A549 cell line showed a higher expression of p-EGFR than the AGS cell line.

### 3.1. p-EGFR Expression in Spheroids

Figure 4 shows changes in p-EGFR expression levels over time. Figure 4a shows control images (no drug) according to days. As shown in the visualized images, cells grew for up to 14 days to form spheroids in 0.5% w/v alginate. Western blotting showed that the relative expression levels of p-EGFR in the spheroid model were higher than in the single-cell model, as shown in Figure 4b. After day seven, spheroids did not continue growing to form big spheroids (cancer spheroid), as shown in Figure 4c. Single-cell or small spheroids grown for under seven days showed low relative expression of p-EGFR. Relative expression was calculated by dividing the area of p-EGFR staining (green) by the area of nucleus staining (blue). At all times before the seven days, p-EGFR expression was weak, and the green fluorescence was faint, as shown in Figure 4a. However, when the spheroids were fully grown over 100 µm in diameter, all cancer spheroids in the alginate spot showed overexpression of p-EGFR (Figure 4a). Figure 4d shows that the relative expression levels of p-EGFR were maintained above 90% after seven days. To confirm whether the drugs inhibit p-EGFR, thus killing the cancer cells in the spheroid model, cancer spheroids overexpressing p-EGFR should be exposed to the drugs after seven days. Hence, we decided on drug treatment for seven days in the spheroid model.

### 3.2. Drug Selection of Targeting p-EGFR Based on the Spheroid Model

Inhibition of p-EGFR occurred early on after administering the drug treatment; immunostaining of the spheroids was conducted at 6, 24, 48, and 144 h after the treatment. Number of cells were effect to whole p-EGFR in an alginate spot. Cell death or cell growth inhibition due to drug may reduce p-EGFR expression compared to that in the control (no drug), even if the drug did not inhibit p-EGFR. Thus, the nucleus of the spheroids should be considered for calculating the relative expression of p-EGFR. After screening 70 drugs (Table 1), cabozantinib was selected as a representative tyrosine phosphorylation inhibitor. Figure 5 shows that cabozantinib inhibited p-EGFR expression. Figure 5a and 5b show that a 36 by 12 spheroid array was formed on the micropillar and that the same micropillar chip was stained with three colors. If the normalized expression of p-EGFR with the drug treatment was significantly different from that in the control (P-value <0.05) and the cytoskeleton was present at >50%, the drugs were classified as p-EGFR inhibitors; thus, cabozantinib was considered an efficient p-EGFR inhibitor in the 3D spheroid model (Figure 5c). If the cytoskeleton was less than 50% after staining (6, 24, 48, and 144 h), normalized p-EGFR was low because the cell did not maintain its structure, and p-EGFR binding to the cell membrane was also disrupted. The cell viability was also measured by staining the spheroid with Calcein AM (Green) (Figure 5d) and used for selecting drugs with p-EGFR expression.

To confirm that the drug inhibited p-EGFR, p-EGFR expression level was observed at 6, 24, 48, and 144 h after drug treatment. Cell viability was also measured six days after drug treatment to determine whether inhibition of p-EGFR caused cell death. Thus, inhibition of p-EGFR and cell viability were essential for identifying the best p-EGFR-targeting drugs. Figure 6 shows the minimum p-EGFR level and cell viability at day seven. The spheroid model showed sensitivity to two drugs (AEE788 (#2), Afatinib (#3)) from among 10 drugs (#2~#11) targeting EGFR out of seventy drugs, whereas the single cell model showed sensitivity to five drugs, as shown in Figure 6a. In the spheroid model, eight drugs targeting EGFR did not evoke a response, which means that the spheroid models show higher resistance to drugs than the single-cell model. Among the 70 drugs, non-EGFR targeting drugs were also screened in the chips. Figure 6b and 6c show the minimum p-EGFR expression and sevenday cell viability of seventy drugs using the singe-cell and spheroid models, respectively. Two non-EGFR targeting drugs (XL147 (#14) and Dovitinib (#39)) showed p-EGFR inhibitor activity and induced cell death in the A549 cancer spheroids as well as the single cells. Figure 6d shows the response change for thirteen drugs in the spheroid model. Thirteen drugs inhibited more than 50% of p-EGFR and induced more than 50% cell death in single-cell models. Cabozantinib (#30), dasatinib (#41), and foretinib (#31) showed similar p-EGFR inhibition, but the spheroids showed high resistance to these drugs. AEE788 (#2), afatinib (#3), regorafenib (#28), bosutinib (#40) did not highly inhibit p-EGFR (phospho Y1092); however, they killed cells in the spheroid model. Tivozanib (#27), AUY922 (#49), dabrafenib (#50), ruxolitinib (#58), and vemurafenib (#60) showed high resistance in the spheroid model. Sixteen drugs inhibited less than 50% p-EGFR expression and induced more than 50% cell death in the single-cell model, but did not induce cell death in the spheroid model (Figure 6e). Although the inhibition of p-EGFR expression was similar, spheroids showed high resistance (high viability) to dacomitinib (#6), Gefitinib (#7), CO-1686 (#11), AZD5363 (#20), Pazopanib (#24), Vandetanib (#29), LY2835219 (#36), AZD4547 (#37), and trametinib (#44) as shown in Figure 6e. Bortezomib (#45) AZD6244 (#43) and AZD2014 (#16) decreased p-EGFR inhibition but still induced cell death. Gefitinib (#7), CO-1686 (#11), and dacomitinib (#6) did not induce cell death and did not inhibit p-EGFR in the spheroid model. Interestingly, in the spheroid model, XL147 (#14), BEZ235 (#19) inhibited the expression of p-EGFR more than in the single-cell model. Overall, the number of drugs targeting p-EGFR was two (Dovitinib and XL147) according to the spheroid model, which is less than the twenty-nine drugs identified from the single-cell model. Thus, cancer spheroid models can be used to narrow down options and identify highly effective target drugs.

## 4. Conclusions

Our cancer spheroid array chip was developed using a micropillar and microwell structure designed to evaluate drug efficacy. To form a high-density cancer spheroid array, the encapsulated A549 cells in alginate were grown for over seven days. Cancer spheroids attached to the micropillar were moved to fresh media or staining reagents by placing them in new microwells filled with new reagents. After forming spheroids with diameter greater than 100 µm in a 12 by 36 pillar array chip (25 mm by 75 mm), we confirmed that the A549 cell line showed overexpression of p-EGFR in cancer spheroids. Cancer spheroids were treated with seventy drugs (six replicates) for evaluating drug efficacy. In the single-cell model, eleven drugs were identified as p-EGFR-targeting drugs, but in the cancer spheroid model, only two drugs were identified as highly efficient p-EGFR-targeting drugs. XL147 and Dovitinib showed p-EGFR inhibition and induced the death of A549 cancer spheroids as well as single cells. After comparing the drug response of single-cell and cancer spheroid models, it was shown that the spheroid model could narrow down the list of drug candidates by identifying high-efficacy p-EGFR-targeting drugs.

Cancer spheroid array chips, by enabling the identification of effective drugs from a huge library, could be useful tools for drug discovery. When compared to the 2D cell culture system, cancer spheroids showed higher resistance to the compounds. Besides, immunostaining of cancer spheroid array chips can serve in determining the mechanism of action of drugs in in vivo-like environments. Because animal and clinical trials for many drug candidates are expensive, having a method to pinpoint the best drug candidate can significantly reduce drug development costs and time.

## Figures and Tables

**Figure 1 micromachines-10-00688-f001:**
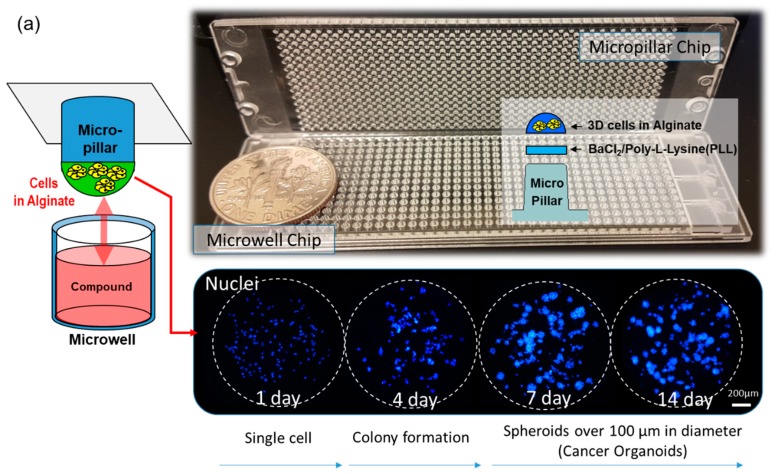
Cancer spheroid array chip based on micropillar and microwell chips. (**a**) Blue represents the nucleus. Spheroids over 100 µm in diameter were formed after seven days. (**b**) Sixteen cell lines (including A549) were grown for seven days in alginate spots on the micropillar chip.

**Figure 2 micromachines-10-00688-f002:**
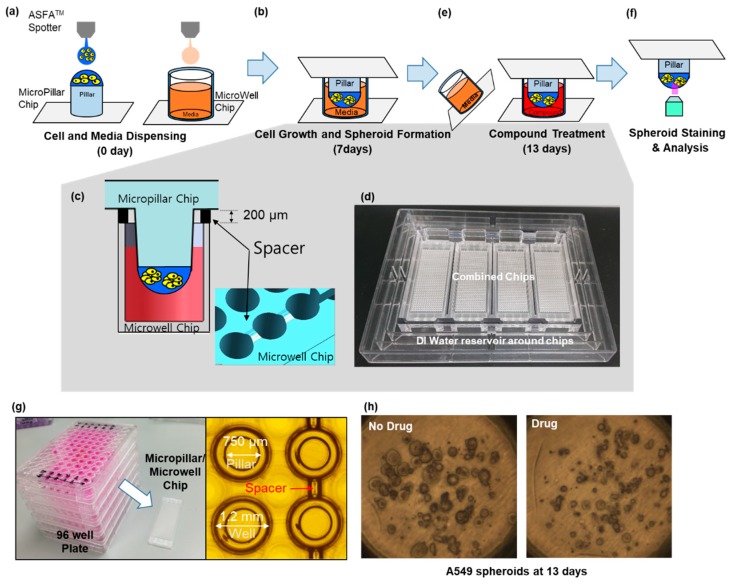
Experimental procedure for the micropillar/microwell chip system. (**a**) Cells and media were dispensed on a micropillar and in a microwell, respectively. (**b**) Cells were immobilized in alginate at the top of the micropillars and dipped into the microwells containing growth media for seven days to form spheroids with diameter over 100 µm. (**c**) The microchip was separated from other microwell chips by a spacer to unify penetration of CO_2_ into the chips. (**d**) To prevent evaporation during incubation, the incubation chamber surrounds the four chips in water. (**e**) Compounds are dispensed into the microwells, and spheroids are exposed to the compounds by moving the micropillar chip to a new microwell chip. (**f**) Spheroids are stained with Calcein-AM, and the dried alginate spots on the micropillar chip are scanned for data analysis. (**g**) Comparison of the combined micropillar/microwell chip with a conventional 96-well plate. (**h**) Cancer spheroid images with and without drug treatment on day 13.

**Figure 3 micromachines-10-00688-f003:**
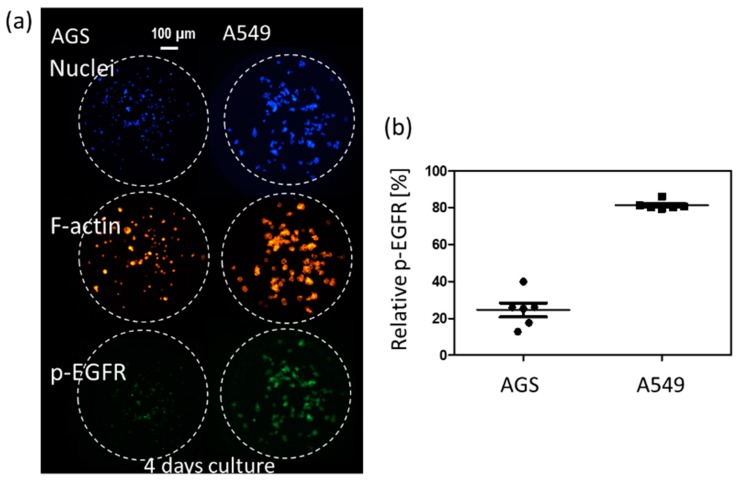
Relative phospho-epidermal growth factor receptor (p-EGFR) in AGS and A549 cell lines in a four day-culture. (**a**) Immunostaining images of two cell line. (**b**) Relative p-EGFR in AGS and A549 cell lines.

**Figure 4 micromachines-10-00688-f004:**
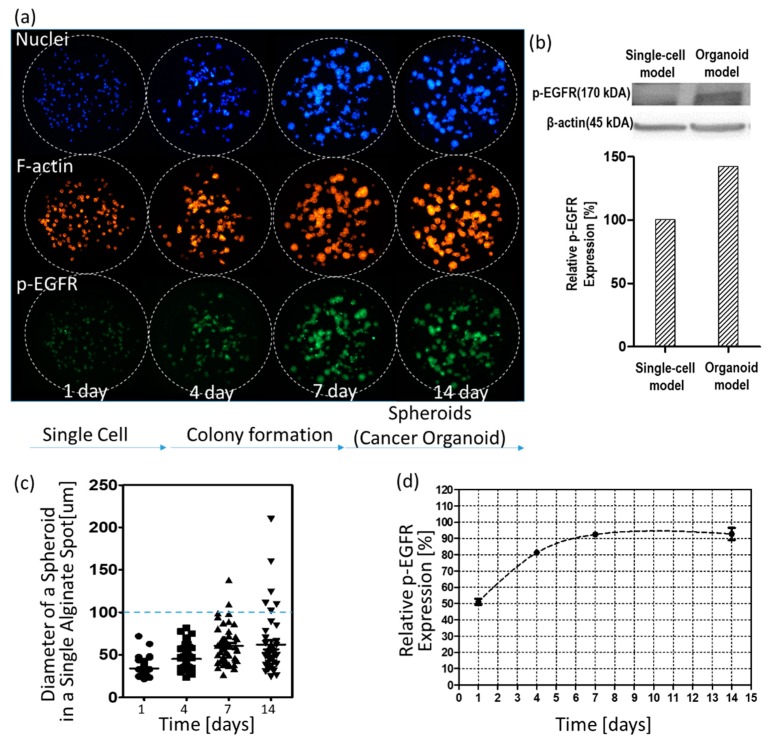
Changes in the nucleus, filamentous actin (F-actin), and p-EGFR in cells and spheroids over time. (**a**) Three color images of the same alginate spots. (**b**) Western blot of p-EGFR expression in single-cell and spheroid models. (**c**) Calculated diameters of spheroids in a single alginate spot over time. (**d**) Relative expression levels of p-EGFR in the spheroid model over time.

**Figure 5 micromachines-10-00688-f005:**
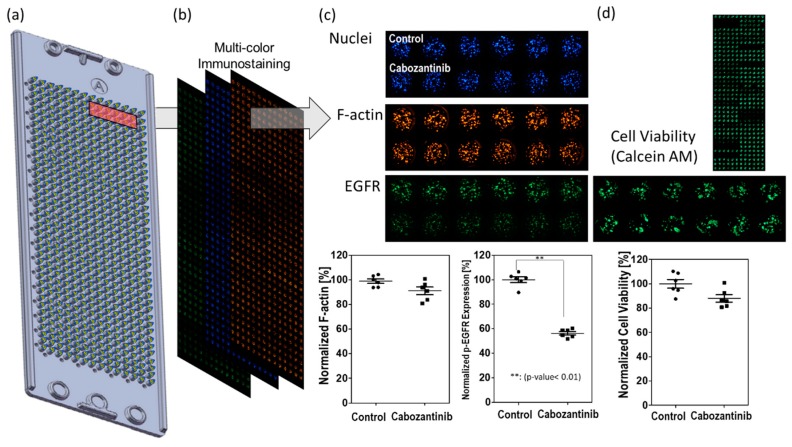
Measurement of p-EGFR inhibition. (**a**) Micropillar chips containing spheroids on the micropillar. (**b**) Three color images were taken of the same micropillar chip. (**c**) Normalized F-actin for detecting the cytoskeleton and normalized expression of p-EGFR for detecting p-EGFR expression. If drug-induced inhibition of normalized p-EGFR is significantly different from that in the control (P-value < 0.05) but not different from that in samples with normalized F-actin, the drugs were classified as p-EGFR inhibitors. The graph shows effective inhibition of p-EGFR on treatment with cabozantinib. (**d**) Cell viability by staining with Calcein AM (Green).

**Figure 6 micromachines-10-00688-f006:**
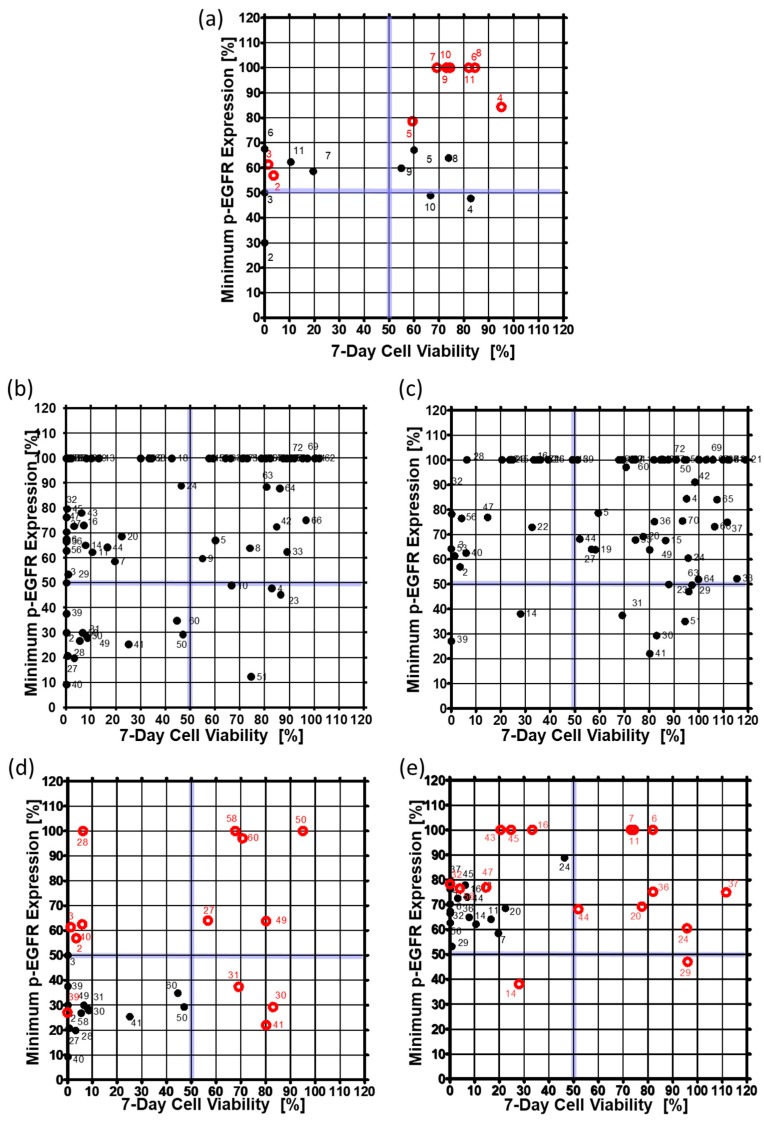
Minimum expression levels of p-EGFR and cell viability on day seven in the single-cell model and the spheroid model. The minimum expression levels of p-EGFR were selected from among the values at 6, 24, 48, and 144 h after drug treatment. (**a**) Expression of p-EGFR and cell viability in the single-cell model. (**b**) Expression of p-EGFR and cell viability in the spheroid model. (**c**) Changes in the expression levels of p-EGFR and cell viability after treatment with thirteen drugs in the single-cell and the spheroid models. Thirteen drugs are under the 50% minimum expression level of p-EGFR and 50% cell viability in the single-cell model. (**d**) Changes in expression of p-EGFR and cell viability due to treatment with sixteen drugs at single-cell and spheroid models. Sixteen drugs were under the 90% minimum expression level of p-EGFR and 50% cell viability in the single-cell model. (**e**) Changes in expression of p-EGFR and cell viability due to treatment with sixteen drugs at single-cell and spheroid models. Sixteen drugs were under the 90% minimum expression level of p-EGFR and 50% cell viability in the single-cell model.

**Table 1 micromachines-10-00688-t001:** p-EGFR expression and relative cell viabilities when the single-cell and spheroids models were exposed to the drugs for seven days.

Drug	Target	Single-Cell Model				Cancer Spheroid Model				Drug	Target	Single-Cell Model				Cancer Spheroid Model			
		p-EGFR Expression [%]		Cell Viability [%]		p-EGFR Expression [%]		Cell Viability [%]				p-EGFR Expression [%]		Cell Viability [%]		p-EGFR Expression [%]		Cell Viability [%]	
		Average	SD	Average	SD	Average	SD	Average	SD			Average	SD	Average	SD	Average	SD	Average	SD
1_DMSO	-	100.0	0.0	100.2	6.5	100.0	0.0	100.0	8.8	37_AZD4547	FGFR1/2/3	76.4	9.5	0.0	0.0	74.9	22.6	111.6	20.6
2_AEE788	EGFR	30.0	8.5	0.0	0.1	56.9	16.1	3.6	1.6	38_BGJ398	FGFR1/2/3	100.0	0.0	66.2	12.7	100.0	0.0	109.8	11.7
3_Afatinib	EGFR	50.0	8.4	0.0	0.0	61.3	20.9	1.4	2.3	39_Dovitinib	Flt3, c-Kit, FGFR1/3, VEGFR1/2/3, PDGFRα/β	37.6	6.9	0.0	0.0	27.1	3.6	0.1	0.1
4_BMS-599626	EGFR	47.7	8.1	82.8	7.6	84.3	6.3	95.1	15.1	40_Bosutinib	dual Src/Abl	9.2	3.2	0.0	0.0	62.5	8.9	6.0	3.1
5_Erlotinib HCl	HER1/EGFR	67.1	12.7	60.0	10.1	78.6	3.9	59.4	7.1	41_Dasatinib	Bcr-Abl	25.4	6.8	25.0	11.1	22.0	2.6	80.3	8.0
6_Dacomitinib	EGFR	67.5	16.0	0.0	0.0	100.0	0.0	82.0	9.2	42_Nilotinib	Bcr-Abl	72.5	13.8	84.8	9.5	91.1	5.8	98.5	10.6
7_Gefitinib	EGFR	58.6	6.0	19.5	19.3	100.0	0.0	73.2	13.5	43_AZD6244	MEK1	78.0	20.6	6.1	1.4	100.0	0.0	20.6	6.3
8_Lapatinib	EGFR	63.9	12.3	73.9	5.8	100.0	0.0	84.5	8.5	44_Trametinib	MEK1/2	64.3	11.7	16.5	5.4	68.1	32.0	52.0	5.4
9_Neratinib	EGFR	59.8	7.2	54.9	5.0	100.0	0.0	73.1	6.0	45_Bortezomib	Proteasome	79.7	12.1	0.3	0.3	100.0	0.0	24.8	11.7
10_CI-1033	EGFR, HER2	48.9	12.9	66.6	6.8	100.0	0.0	69.2	10.7	46_Carfilzomib	Proteasome	100.0	0.0	0.2	0.2	100.0	0.0	39.2	9.5
11_CO-1686	EGFR	62.3	14.3	10.5	10.7	100.0	0.0	74.4	12.5	47_ABT-199	Bcl-2	72.7	8.7	3.1	0.9	76.8	13.4	14.7	3.8
12_BKM120	PI3K	100.0	0.0	1.9	0.8	100.0	0.0	36.1	4.4	48_ABT-888	PARP	100.0	0.0	82.2	5.4	100.0	0.0	112.3	8.7
13_BYL719	PI3K	100.0	0.0	13.0	4.2	100.0	0.0	49.0	9.1	49_AUY922	HSP (e.g. HSP90)	27.8	5.5	8.5	2.7	63.8	12.4	80.2	11.7
14_XL147	PI3K	65.1	11.1	7.7	2.1	38.0	8.6	28.0	3.8	50_Dabrafenib	BRAFV600	29.3	11.0	47.0	10.1	100.0	0.0	95.1	6.8
15_Everolimus	mTOR	100.0	0.0	57.5	4.0	67.6	16.3	86.6	12.1	51_Ibrutinib	Btk, modestly potent to Bmx, CSK, FGR, BRK, HCK	12.4	5.6	74.5	9.4	35.0	2.8	94.5	16.8
16_AZD2014	mTOR	73.1	9.6	6.9	2.1	100.0	0.0	33.4	7.6	52_LDE225	Smoothened	100.0	0.0	90.3	12.1	100.0	0.0	94.3	15.9
17_PF-05212384	P3k/mTOR	100.0	0.0	1.6	0.6	100.0	0.0	34.9	5.9	53_LDK378	ALK	100.0	0.0	0.0	0.0	64.2	19.9	0.0	0.0
18_XL765	P3k/mTOR	100.0	0.0	42.5	3.8	100.0	0.0	68.7	3.0	54_LGK-974	PORCN	100.0	0.0	72.8	4.5	100.0	0.0	81.9	7.3
19_BEZ235	P3k/mTOR	100.0	0.0	10.0	3.4	63.8	4.8	58.3	7.7	55_Olaparib	PARP1/2	100.0	0.0	59.2	5.3	67.8	18.0	74.5	17.3
20_AZD5363	Akt1/2/3	68.7	7.5	22.4	8.0	69.2	5.5	77.6	9.0	56_Panobinostat	HDAC	62.8	8.6	0.1	0.1	76.4	2.5	4.2	1.3
21_Axitinib	VEGFR1/2/3, PDGFRβ and c-Kit	100.0	0.0	34.4	12.0	100.0	0.0	119.0	9.6	57_PF-04449913	HSP90	100.0	0.0	80.6	15.1	100.0	9.2	87.8	15.2
22_Cediranib	VEGFR, Flt	100.0	0.0	0.0	0.0	72.9	14.9	32.7	44.7	58_Ruxolitinib	JAK1/2	29.9	12.2	6.5	2.3	100.0	0.0	67.8	8.1
23_Imatinib	v-Abl, c-Kit and PDGFR	45.3	9.3	86.4	3.7	49.9	10.9	87.9	7.8	59_Sotrastaurin	PKC	100.0	0.0	30.0	8.3	100.0	0.0	51.0	14.9
24_Pazopanib HCl	VEGFR1/2/3, PDGFR, FGFR, c-Kit	89.0	14.2	46.2	11.5	60.4	23.1	95.8	8.5	60_Vemurafenib	B-RafV600E	34.8	5.5	44.5	2.8	97.1	5.8	70.7	24.4
25_Sunitinib	VEGFR2 and PDGFRβ	100.0	0.0	0.0	0.0	100.0	0.0	39.3	43.6	61_Vismodegib	Hedgehog/smothen	100.0	0.0	97.2	9.7	100.0	0.0	111.9	5.6
26_Tandutinib	FLT3, PDGFR, and KIT	100.0	0.0	8.0	5.0	100.0	0.0	23.6	4.2	62_PHA-665752	c-Met inhibitor	100.0	0.0	102.0	11.4	100.0	0.0	130.0	13.2
27_Tivozanib	VEGFR, c-Kit, PDGFR	19.8	4.6	3.2	2.2	64.0	22.2	56.7	5.9	63_TMZ	alkylating agent	88.5	6.3	80.7	7.8	49.6	21.1	97.3	13.2
28_Regorafenib	VEGFR1/2/3, PDGFRβ, Kit, RET and Raf-1	20.7	4.8	0.6	0.7	100.0	0.0	6.3	1.1	64_Amoral	morpholine antifungal drug	87.9	16.6	86.0	9.5	51.9	22.1	99.9	9.5
29_Vandetanib	VEGFR2	53.3	4.7	0.7	1.1	47.0	23.6	96.0	14.6	65_Mevas	HMG-CoA reductase inhibitor	100.0	0.0	88.9	7.2	84.0	11.4	107.4	8.1
30_Cabozantinib	VEGFR2,c-Met, Ret, Kit, Flt-1/3/4, Tie2, and AXL	28.7	7.0	8.0	5.1	29.3	13.3	83.0	16.8	66_Amio	antiarrhythmic medication	75.2	27.9	96.6	3.9	73.1	10.7	106.5	2.5
31_Foretinib	HGFR and VEGFR, mostly for Met and KDR	26.8	6.5	5.4	3.3	37.4	3.3	69.1	19.1	67_Flu	Anticholesterol agent. HMG-CoA inhibitor	100.0	0.0	87.4	15.2	100.0	0.0	105.7	5.3
32_Crizotinib	Met, ALK	70.4	4.4	0.0	0.0	78.1	23.7	0.2	0.2	68_Myco_acid	Inosine-5’-monophosphate dehydrogenase inhibitor	100.0	0.0	33.5	8.7	100.0	0.0	100.3	4.8
33_INCB28060	Met	62.5	13.5	88.9	7.4	52.2	10.6	115.4	12.9	69_Raloxi	Estrogen receptor inhibitor	100.0	0.0	95.1	4.9	100.0	0.0	103.0	13.4
34_LEE011	CDK4/6	100.0	0.0	64.2	9.0	100.0	0.0	86.0	5.7	70_Astemi	Histamine receptor ligand	100.0	0.0	78.6	8.5	75.4	14.9	93.5	11.4
35_PD 0332991	CDK4/6	100.0	0.0	1.2	0.9	100.0	0.0	85.7	10.1	71_Ferre	Retinoic acid receptor ligand	100.0	0.0	71.0	7.4	100.0	0.0	111.9	16.6
36_LY2835219	CDK4/6	66.6	8.5	0.0	0.0	75.1	25.9	82.1	11.2	-	-	-	-	-	-	-	-	-	-

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
