# Peer review of "A Cancer Spheroid Array Chip for Selecting Effective Drug"

_micromachines, 2019, doi:10.3390/mi10100688_

Round 1

Reviewer 1 Report

Lee et al. present a cancer organoid array chip for drug screening. The device was based on their previously developed micropillar and microwell chip platforms. They modified the design to allow for long-term 3D tissue culture (up to 14 days) by introducing a 0.2 mm spacer (thicker than that in their previous paper, Lee et al, Anal. Chem. 2018) between the miropillar and the microwell chips for better CO2 penetration and a DI water reservoir for minimized medium evaporation. The authors were able to utilize this modified platform to generate cancer organoids (“mature spheroids”) and screen drugs in a high throughput manner. The major issue is that the drug screening experiment they designed and conducted does not support their conclusion, “the organoid models could identify highly potent drugs targeting p-EGFR more effectively than the single-cell model, which narrows down the option of potential drug candidates”. Details are discussed below.

Specific points

The cancer organoid model needs validation. To which extent can the model predict cancer tissue drug response in vivo? They did a comparison on drug responses between a “single-cell” model and the cancer organoid model. The “sing-cell” model itself (where cells are in alginate gel as single cells or small spheroids) has not been well characterized in terms of its performance in prediciting clinical drug responses. Testing some model drugs with known clinical responses can help validate the current organoid model. The comparison between the single also does not support their conclusion that “the organoid models could identify highly potent drugs targeting p-EGFR more effectively than the single-cell model”. The quality of a drug screening study is not evaluated by the final number of drugs selected, but by its accuracy in predicting clinical drug responses. Clearly the two models showed very different drug responses in terms of p-EGFT expression and cell viability. Which model is more accurate in recapitulating in vivo cancer drug responses? While the cancer organoids model narrowed down the options of drug candidates, it is not clear how this short list aligns with actual top candidates. Without confirming the accuracy of the cancer organoid model based drug screening, it is hard to conclude that drug screening based on this model is more effective. The authors defined cancer organoids as mature spheroids yet without clearly defining “mature” spheroids. They referred to spheroids as mature ones after they grew for 7 day in 3D gel and become larger than 100um. Why did the authors pick 7 days in culture or 100 um in diameter as a cutoff for mature spheroids? The spheroids in Figure 2h (no Drug) seem not very uniform in size. What does the spheroid size distribution look like after 1-2 weeks culture in the alginate gel? Will the variation in size affect their “maturity” thus affect their drug responses? It not explicitly stated why they compared AGS and A549 two cell lines. What was the purpose to pretest p-EGFR in AGS? The authors mentioned that “Day-to-day or chip-to-chip variations of fluorescence intensity during immunostaining were quite high” and “the fluorescence intensity of p-EGFR was not absolute.” (P6) Were those p-EGFR fluorescence images showing absolute intensity? how to compare p-EGFR fluorescence images on different days (such as figure 4a), which assumably were not normalized images? In the introduction (P2, line 52, “forming cancer organoids in a high-throughput manner is technically difficult”. There have been many technologies developed for morning cancer organoids, such as hang drop technology, agarose microwells etc. Page 1, line 32. “when cancer cells become attached to a plastic dish, they grow into a single layer, …” This is not necessarily true, and likely cancer cell type dependent. Revise the sentence to be more accurate. Page 1, line 45. “A scaffold-free model allows cells to grow together without an extra-cellular matrix (ECM) and …” cells can form their own ECM during culture. “Without exogenous ECM” would be more accurate. Fig 1b. showed 16 cancer cell lines. But there is no source information as well as methods regarding these 16 cell lines except for A549 and AGS. Calcein AM staining concentration for cell viability is not described. Only stock concentration was mentioned. P5 line159, “3D cultured cells more resistance to the drugs”. Fix the sentence first. Any reference? This is likely cell type and drug dependent. Even based on the results from the current manuscript, this statement is not always true. 20uM was used for all drugs. But clearly drug responses are likely dose dependent. The authors need to justify the use of a single dose for all drugs and the choice of 20uM. The manuscript needs polishing. There are also many grammatical errors.

Author Response

We attached file.

Reviewer 2 Report

I think the device design described in this paper is neat and effective. The biological experiments to demonstrate the application of the device are extensive. I only have two minor suggestions for the authors, before this paper can be accepted for publication.

The image quality of Figure 1 is too low. Please find a way to reserve high resolution in the document. I feel this cancer model is more like a "spheroid" rather than an "organoid," because cancer cells cannot constitute an organ-like model and this study shows no study on this model's organ-like function.

Author Response

We attached file.

Round 2

Reviewer 1 Report

Reviewer 1 Round 2 comments (in green)

Lee et al. present a cancer organoid array chip for drug screening. The device was based on their previously developed micropillar and microwell chip platforms. They modified the design to allow for long-term 3D tissue culture (up to 14 days) by introducing a 0.2 mm spacer (thicker than that in their previous paper, Lee et al, Anal. Chem. 2018) between the miropillar and the microwell chips for better CO2 penetration and a DI water reservoir for minimized medium evaporation. The authors were able to utilize this modified platform to generate cancer organoids (“mature spheroids”) and screen drugs in a high throughput manner. The major issue is that the drug screening experiment they designed and conducted does not support their conclusion, “the organoid models could identify highly potent drugs targeting p-EGFR more effectively than the single-cell model, which narrows down the option of potential drug candidates”. Details are discussed below.

A new author was added as the first author, but the manuscript seems only had some minor changes. The “author contributions” section did not state the contribution of the new author. The authorship should reflect one’s contribution. 

The cancer organoid model needs validation. To which extent can the model predict cancer tissue drug response in vivo? They did a comparison on drug responses between a “single-cell” model and the cancer organoid model. The “sing-cell” model itself (where cells are in alginate gel as single cells or small spheroids) has not been well characterized in terms of its performance in prediciting clinical drug responses. Testing some model drugs with known clinical responses can help validate the current organoid model. The comparison between the single also does not support their conclusion that “the organoid models could identify highly potent drugs targeting p-EGFR more effectively than the single-cell model”. The quality of a drug screening study is not evaluated by the final number of drugs selected, but by its accuracy in predicting clinical drug responses. Clearly the two models showed very different drug responses in terms of p-EGFT expression and cell viability. Which model is more accurate in recapitulating in vivo cancer drug responses? While the cancer organoids model narrowed down the options of drug candidates, it is not clear how this short list aligns with actual top candidates. Without confirming the accuracy of the cancer organoid model based drug screening, it is hard to conclude that drug screening based on this model is more effective.

Response) This paper focused on a system including the optimized chips and accessory (incubation chamber) for culturing organoid with 3D culture method. Our group were not experts about in-vivo test and drugs. Thus, we select “Micromachines” related to tools to culturing organoids. Unfortunately, we could not provide in vivo data and clinical drug response. So, as reviewer, comments, we change our conclusion “The organoid model is more resistant to drugs targeting p-EGFR than the single cell model., which narrows down the option of drug candidates.” We eliminate confused sentence like “highly potent drugs identify”

Round 2 comment: first, “The organoid model is more resistant to drugs targeting p-EGFR than the single cell model” not true for some drugs tested in the current study. Secondly, the review was not asking for in vivo or clinical data. While the multipillar and microwell chip platform is unique and useful, the platform itself had been published several times. The current device only made very subtle changes to the original design. The adaptation made in this manuscript was to support long term culture. If the focus was on the system for culturing organoid with 3D culture method, as the authors responded, then they need to emphasize on how the system supported the cancer spheroid culture and characterize the “mature spheroid”. But they actually claimed a lot more than that. They tried to demonstrate the cancer array chip as a useful platform for drug screening. Now the questions arise. Can this be used for screening for effective drugs? Why is this system better than the single cell model? There is simply lack of validation of the model. Why the 7d spheroids are better than 1d single cells?  Testing some model drugs with known clinical responses can help validate the current organoid model. They need demonstrate the drug responses from the current model align well with the clinical responses of some drugs (pick some model drugs with known clinical response for validation). It is hard to derive any meaningful conclusion using an unvalidated platform to test new drugs.

The authors defined cancer organoids as mature spheroids yet without clearly defining “mature” spheroids. They referred to spheroids as mature ones after they grew for 7 day in 3D gel and become larger than 100um. Why did the authors pick 7 days in culture or 100 um in diameter as a cutoff for mature spheroids? The spheroids in Figure 2h (no Drug) seem not very uniform in size. What does the spheroid size distribution look like after 1-2 weeks culture in the alginate gel? Will the variation in size affect their “maturity” thus affect their drug responses?

Response) In single alginate spot, standard deviation of spheroid size is about 20~60% because alginate had different size of pore and some nearby cells migrated and grow together. We change Fig.4c to show size of individual spheroid size. As shown in Fig.4c, after day 7, 100 um-diameter spheroids are formed in alginate spot. In single spot, spheroid size variation is large. However, spot-to-spot variation of total spheroid area is under 25% at day 7 and 14. We revised Fig.4c. and caption.

Still, define “mature spheroid”?

It not explicitly stated why they compared AGS and A549 two cell lines. What was the purpose to pretest p-EGFR in AGS?

Response) Iimmunostaining of p-EGFR using chip needs to verify by measuring p-EGFR in AGS and A549 which are well-known cell lines of p-EGFR overexpression. In this experiment, A549 had higher p-EGFR expression than AGS and we selected A549 cell line to screen drugs.

The authors mentioned that “Day-to-day or chip-to-chip variations of fluorescence intensity during immunostaining were quite high” and “the fluorescence intensity of p-EGFR was not absolute.” (P6) Were those p-EGFR fluorescence images showing absolute intensity? how to compare p-EGFR fluorescence images on different days (such as figure 4a), which assumably were not normalized images?

Response) We remove confused sentence “the fluorescence intensity of p-EGFR was not absolute”. The expression of p-EGFR after drug treatment should be compared with the control (no drugs) in the same chip. Thus, we normalized p-EGFR by dividing relative expression of p-EGFR found in the control sample with that of one of the drug treatments on the same chip.

Round 2 comment: they authors did not answer the reviewer’s question. yes, quantitatively, relative expression of p-EGFP can be used to exam drug response. Yet, the question was regarding the comparison of fluorescence images on different days (eg. Figure 4a). The normalization can only be made within the same chip on the same day, since “Day-to-day or chip-to-chip variations of fluorescence intensity during immunostaining were quite high” even with the same staining protocol. So, if the fluorescence intensity on different dates is not correlated with the expression level, figures like 4a can be misleading.

In the introduction (P2, line 52, “forming cancer organoids in a high-throughput manner is technically difficult”. There have been many technologies developed for morning cancer organoids, such as hang drop technology, agarose microwells etc. Page 1, line 32.

Response) We revised sentence. “Thus, there have been many technologies developed, such as hang drop technology, agarose microwells, microfluidic chips, etc.”

Round 2 comment: While the authors basically took the reviewer’s sentence, but at least they need to add on relevant references. There is a lack of review on the current available technologies and the existing issues. The following paragraph starts with “to overcome this problem,…” Not sure what “this problem ” refers to.

“when cancer cells become attached to a plastic dish, they grow into a single layer, …” This is not necessarily true, and likely cancer cell type dependent. Revise the sentence to be more accurate. Page 1, line 45.

Response) we revised confused sentence like “However, when cancer cells are cultured in plastic dishes, a single layer or colony is formed depending on the cell type”

Round 2 comment: Could be multilayers, not necessarily a single layer or colonies.

“A scaffold-free model allows cells to grow together without an extra-cellular matrix (ECM) and …” cells can form their own ECM during culture. “Without exogenous ECM” would be more accurate.

Response) we revised confused sentence like A scaffold-free model allows cells to grow together without exogenous extra-cellular matrix (ECM)

Fig 1b. showed 16 cancer cell lines. But there is no source information as well as methods regarding these 16 cell lines except for A549 and AGS.

Response) We add source information in line 99.

Calcein AM staining concentration for cell viability is not described. Only stock concentration was mentioned.

Response) Staining solution was made by adding 1.0 µL of calcein AM (4 mM stock from Invitrogen) in 8 mL of 140 mM NaCl supplemented with 20 mM CaCl2. We add this sentence in line 197.

P5 line159, “3D cultured cells more resistance to the drugs”. Fix the sentence first. Any reference? This is likely cell type and drug dependent. Even based on the results from the current manuscript, this statement is not always true. 20uM was used for all drugs. But clearly drug responses are likely dose dependent. The authors need to justify the use of a single dose for all drugs and the choice of 20uM.

Response) We fixed confused sentence and add reference. The sentence is changed like “In previous our study [24, 25], most of drugs showed high resistance in 3D cultured cells.”. Actually, we choice of 20 uM based on our previous experiments. To analysis 70 drugs in single chip, we could not apply multi-dose.

In our previous studies ….

A brief reasoning of the choice of 20um should be mentioned in the manuscript.

The manuscript needs polishing. There are also many grammatical errors.

Response) We check grammatical errors with English Expert.

Checked or will check?

Author Response

We attach file

Round 3

Reviewer 1 Report

This revision made improvement. Some minor changes are needed.

“when cancer cells become attached to a plastic dish, they grow into a single layer, …” This is not necessarily true, and likely cancer cell type dependent. Revise the sentence to be more accurate. Page 1, line 45.

Response) we revised confused sentence like “However, when cancer cells are cultured in plastic dishes, a single layer or colony is formed depending on the cell type”

Round 2 comment: Could be multilayers, not necessarily a single layer or colonies.

Round 2 Response) We revised like “when cancer cells are cultured in plastic dishes, a multilayer is formed depending on the cell type”.

Round 3 comment: not sure the authors understood the reviewer's comments. Can they read the previous comments regarding this sentence carefully and revise to make it more accurate?

-

The authors mentioned that “Day-to-day or chip-to-chip variations of fluorescence intensity during immunostaining were quite high” and “the fluorescence intensity of p-EGFR was not absolute.” (P6) Were those p-EGFR fluorescence images showing absolute intensity? how to compare p-EGFR fluorescence images on different days (such as figure 4a), which assumably were not normalized images?

Response) We remove confused sentence “the fluorescence intensity of p-EGFR was not absolute”. The expression of p-EGFR after drug treatment should be compared with the control (no drugs) in the same chip. Thus, we normalized p-EGFR by dividing relative expression of p-EGFR found in the control sample with that of one of the drug treatments on the same chip.

Round 2 comment: they authors did not answer the reviewer’s question. yes, quantitatively, relative expression of p-EGFP can be used to exam drug response. Yet, the question was regarding the comparison of fluorescence images on different days (eg. Figure 4a). The normalization can only be made within the same chip on the same day, since “Day-to-day or chip-to-chip variations of fluorescence intensity during immunostaining were quite high” even with the same staining protocol. So, if the fluorescence intensity on different dates is not correlated with the expression level, figures like 4a can be misleading.

Round 2 Response) Ok, we remove confused sentence. We did not use low data of the fluorescence intensity directly. To compare p-EGFR expression and cell viability from different chips, we normalized cell viability and p-EGFR expression by comparing with the control (no drugs) in the same chip. We revised this in line 181.

Round 3 comments.The author deleted the sentence on "day to day" variation. It was not this sentence that caused confusion, if it was the fact. The question was regarding figure 4a, and the authors never answered in both round 1 and round 2 revision.

Author Response

We attach file

This manuscript is a resubmission of an earlier submission. The following is a list of the peer review reports and author responses from that submission.

Round 1

Reviewer 1 Report

The authors describe the use of a micropillar/microwell chip to high throughput creation of tumor organoids created from A549 cells. These organoid arrays are then employed for a multi drug screening study. The work presented has some merit, as figuring out how to do high throughput screening studies using 3D organoid models is incredibly important. However, the impact of the paper is significantly diminished due to the poor grammar and many mistakes in the text. This will require significant editing and proofreading to improve. Moreover, the study also suffers from the use of only a single cell line.

Specific comments:

There are many grammatical mistakes and typos throughout the text, including capitalization errors, incorrect verb tenses, and other problems. Significant edits and proofreading are required.

The authors state multiple times that one of main hurdles with organoids is that changing media often damages cells. Do the authors experience this themselves? There are no references backing up these statements. Changing media and damaging organoids is not a problem that we have encountered in many years of organoid studies.

How the alginate was crosslinked on the pillars is not explained.

Why was f-actin used as a marker for viability? It is not a marker of viability, but rather a structural protein. An assay such as ATP, mitochondrial metabolism, live/dead staining, caspase expression would be much more appropriate.

A comparison cell line with different EGFR expression would be a useful addition.

Discussion of the utility of this system in the future would add to the quality of the paper. How can this platform be useful?

How were the drugs administered to the individual organoids. It would seem that this step, would be low throughput since each individual drug concentration would need to be applied individually.

There is a low number of references despite much recent work on this topic.

Author Response

I attached response file

Reviewer 2 Report

Lee et al. reported in this study an improved method to generate cancer organoid in an array chip for high-throughput anti-cancer drug screening applications. This study follows the current trend in drug research and development using human 3D cell culture techniques to support, replace, or reduce in vivo studies on animals. Building from their previous study, the authors attempted to create 100-μm cancer organoid in their microwell platform and demonstrate its use to identify phospho-EGFR targeting drugs in a high throughput manner. The authors addressed the issue of minimizing damage to organoids during media change and reagent exchange via their micropillar and microwell configuration. While it is an interesting concept and could contribute to the future of drug discovery, it is not technological innovative, and will require some additional work and manuscript revision.

This reviewer has a few comments and suggestions for the authors: 

1. It is widely known that not all cancer cell lines will form spheroid or organoid, or they will behave similarly in 3D culture. For instance, MDA-MB-231, a triple negative breast cancer cell line, does not self-aggregate into spheroids, nor they will ever stay intact in forced spheroid configuration (e.g. encapsulating in hydrogel). In this study, the authors only picked one lung cancer cancer cell line A549 that is well-known to self-assemble into spheroid form. This reviewer would like the authors to comment on whether this array chip platform will only work for cells that can self-assemble into spheroid. Specifically, please describe any additional cancer cell lines that have been tested in this configuration, and how much success can be achieved. If only a subset of cancer cell lines can be used, the authors should address this point in the manuscript.

2. As a follow up suggestion, the authors should include 1-2 more cancer cell lines to this manuscript, even as a proof-of-concept. To make a case for this array chip for high throughput drug screening application, the authors should demonstrate that it can be used with different cancer cell lines. It is generally understood that no system is perfect, as long as the boundary is clearly defined. 

3. The authors should re-organize the Materials & Methods. This reviewer found it is hard to follow the current order. As a suggestion for the order of these sections: Fabrication, Cell Culture, Experimental Procedure, Drug Response Comparison Single Cells vs. Organoid Models, Multi-color Analysis, Western Blot Assay. This order follows the data in the Results section better. 

4. Line 84-85: Please describe how long fixing with 4% PFA is required, and whether fixing duration is dependent on the number of spheroids formed in each micropillar/microwell structure. For example, if there are 100 spheroids in well #1 versus 50 spheroids in well #2, the duration needs to be optimized. Overfixing with 4% PFA can over-crosslink proteins inside cells and effect immunostaining quality. Since analysis of this array chip heavily rely on immunostaining, it is crucial to address the fixation process in details. 

5. For the drug study to evaluate the efficacy of p-EGFR inhibition, please address the rationale to choose the concentration at 20 μM. This concentration is extremely high, especially for tyrosine kinase inhibitors. Newer generation of tyrosine kinase inhibitors are effective at hitting multiple targets such as VEGFR2/EGFR/ etc. at nM concentrations in kinase assays. Even for cell growth assay, IC50 of these drugs on cancer cells are in the range of 1-10 μM. This reviewer is gravely concerned that 20 μM drug concentration chosen in this manuscript actually supports the idea of screening for p-EGFR inhibition, and the sensitivity of this assay.

6. This reviewers is gravely concerned about the method to quantify cell viability solely on F-actin staining and its fluorescent readout. Since these cancer cells formed spheroids, it is more likely the the outer shell cells will be targeted harder than the inner core cells due to diffusion limit. Without knowing exactly how many cells per spheroids before and after drug treatment, or how much the spheroids have shrunken in size, it is almost impossible to determine cell viability in this manner. One could argue that 100 cells from a spheroid are 100% viable via F-actin staining after drug treatment, but this spheroid has 200 cells to begin with, thus 50% of the cells have been killed by this drug. In addition, a fluorescent plate reader cannot achieve precise measurement of fluorescent signal to noise from a lump of many spheroids in a well. This reviewer suggest the authors to choose a different method to quantify cell viability. One possibility is to label these cancer cells with a fluorescent protein via lentivirus, so a fluorescent readout before and after drug treatment can be measured to calculate percent reduction. This still has limit depending on the fluorescent plate reader, but it is a better internal control.

7. The drug response comparison between single cell and organoid cell models is greatly flawed in experimental design. The authors made an assumption that the number of cells seeded on Day 1 (single cell model) is equivalent to the number of cells after 7 days as they assemble into organoids, thus the higher level of p-EGFR in organoid model is due to the difference in structural configuration. However, there is a possibility that the number of cells have significantly increased during that 7-day period while they assemble into spheroid form, thus increasing the total amount of p-EGFR collected for Western blot. One could argue that letting single cells growing for 7 days in a conventional well plate can still achieve the same amount of p-EGFR as organoid model. This reviewer suggests the authors to consider additional control experiments for revision. 

8. With regard to quantifying p-EGFR relative expression, this reviewer is concerned that the authors did not have a baseline level at T=0h for normalization purpose. The authors made the assumption that before drug treatment, all wells (control wells and drug-testing wells) have the same level of p-EGFR, thus any change in p-EGFR readout at different timepoints after is due to drug effect. It is possible that p-EGFR levels are vastly different between wells due to vastly different number of cells per spheroids, and spheroids per well. This reviewer would like the authors to address this concern with additional control experiments.   

9. Minor: Figure 1 - Typo in labeling "Nuclues". Suggesting to revise to "Nuclei" to address multiple cell nuclei in this figure inset.  

Author Response

I attach response file

Round 2

Reviewer 1 Report

The edits made to the manuscript have significantly improved it overall. The grammar is much better, as are several of the results and method descriptions.

There remains one significant problem with the data however. Both reviewers critiqued the use of f-actin staining as a sign of viability. It was suggested to use another output metric such as ATP activity, mitochondrial metabolism, or staining for apoptotic markers. However, no such additions were made. The authors response was simply that they removed the term viability from the text and instead state that f-actin is used to measure the cytoskeleton. This makes little sense an needs to be addressed.